# Placeto: Learning Generalizable Device Placement Algorithms for Distributed Machine Learning

**Ravichandra Addanki**[*], **Shaileshh Bojja Venkatakrishnan**,
**Shreyan Gupta**, **Hongzi Mao**, **Mohammad Alizadeh**

MIT Computer Science and Artificial Intelligence Laboratory

{addanki, bjjvnkt, shreyang, hongzi, alizadeh}@mit.edu

## Abstract

We present Placeto, a reinforcement learning (RL) approach to efficiently find device placements for distributed neural network training. Unlike prior approaches that only find a device placement for a specific computation graph, Placeto can learn generalizable device placement policies that can be applied to *any* graph. We propose two key ideas in our approach: (1) we represent the policy as performing iterative placement improvements, rather than outputting a placement in one shot; (2) we use graph embeddings to capture relevant information about the structure of the computation graph, without relying on node labels for indexing. These ideas allow Placeto to train efficiently and generalize to unseen graphs. Our experiments show that Placeto requires up to $6.1\times$ fewer training steps to find placements that are on par with or better than the best placements found by prior approaches. Moreover, Placeto is able to learn a generalizable placement policy for any given family of graphs, which can then be used without any retraining to predict optimized placements for unseen graphs from the same family. This eliminates the large overhead incurred by prior RL approaches whose lack of generalizability necessitates re-training from scratch every time a new graph is to be placed.

## 1 Introduction & Related Work

The computational requirements for training neural networks have steadily increased in recent years. As a result, a growing number of applications [14, 21] use distributed training environments in which a neural network is split across multiple GPU and CPU devices. A key challenge for distributed training is how to split a large model across multiple heterogeneous devices to achieve the fastest possible training speed. Today device placement is typically left to human experts, but determining an optimal device placement can be very challenging, particularly as neural networks grow in complexity (e.g., networks with many interconnected branches) or approach device memory limits. In shared clusters, the task is made even more challenging due to interference and variability caused by other applications.

Motivated by these challenges, a recent line of work [13, 12, 6] has proposed an automated approach to device placement based on reinforcement learning (RL). In this approach, a neural network policy is trained to optimize the device placement through repeated trials. For example, Mirhoseini et al. [13] use a recurrent neural network (RNN) to process a computation graph and predict a placement for each operation. They show that the RNN, trained to minimize computation time, produces device placements that outperform both human experts and graph partitioning heuristics such as Scotch [18]. Subsequent work [12] improved the scalability of this approach with a hierarchical model and explored more sophisticated policy optimization techniques [6].

---

[*]Corresponding author

Although RL-based device placement is promising, existing approaches have a key drawback: they require significant amount of re-training to find a good placement for each computation graph. For example, Mirhoseini et al. [13] report 12 to 27 hours of training time to find the best device placement for several vision and natural language models; more recently, the same authors report 12.5 GPU-hours of training to find a placement for a neural machine translation (NMT) model [12]. While this overhead may be acceptable in some scenarios (e.g., training a stable model on large amounts of data), it is undesirable in many cases. For example, high device placement overhead is problematic during model development, which can require many ad-hoc model explorations. Also, in a shared, non-stationary environment, it is important to make a placement decision quickly, before the underlying environment changes.

Existing methods have high overhead because they do not learn *generalizable* device placement policies. Instead they optimize the device placement for a *single* computation graph. Indeed, the training process in these methods can be thought of as a search for a good placement for one computation graph, rather than a search for a good placement *policy* for a class of computation graphs. Therefore, for a new computation graph, these methods must train the policy network from scratch. Nothing learned from previous graphs carries over to new graphs, neither to improve placement decisions nor to speed up the search for a good placement.

In this paper, we present Placeto, a reinforcement learning (RL) approach to learn an efficient algorithm for device placement for a given family of computation graphs. Unlike prior work, Placeto is able to transfer a learned placement policy to unseen computation graphs from the same family without requiring any retraining.

Placeto incorporates two key ideas to improve training efficiency and generalizability. First, it models the device placement task as finding a sequence of *iterative placement improvements*. Specifically, Placeto's policy network takes as input a current placement for a computation graph, and one of its node, and it outputs a device for that node. By applying this policy sequentially to all nodes, Placeto is able to iteratively optimize the placement. This placement improvement policy, operating on an explicitly-provided input placement, is simpler to learn than a policy representation that must output a final placement for the entire graph in one step.

Placeto's second idea is a neural network architecture that uses *graph embeddings* [3, 4, 8] to encode the computation graph structure in the placement policy. Unlike prior RNN-based approaches, Placeto's neural network policy does not depend on the sequential order of nodes or an arbitrary labeling of the graph (e.g., to encode adjacency information). Instead it naturally captures graph structure (e.g., parent-child relationships) via iterative message passing computations performed on the graph.

Our experiments show that Placeto learns placement policies that outperform the RNN-based approach over three neural network models: Inception-V3 [23], NASNet [28] and NMT [27]. For example, on the NMT model Placeto finds a placement that runs $16.5\%$ faster than the RNN-based approach. Moreover, it also learns these placement policies substantially faster, with up to $6.1\times$ fewer placement evaluations, than the RNN approach. Given any family of graphs Placeto learns a generalizable placement policy, that can then be used to predict optimized placements for unseen graphs from the same family without any re-training. This avoids the large overheads incurred by RNN-based approaches which must repeat the training from scratch every time a new graph is to be placed.

Concurrently with this work, Paliwal et al. [17] proposed using graph embeddings to learn a generalizable policy for device placement and schedule optimization. However, their approach does not involve optimizing placements directly; instead a genetic search algorithm needs to be run for several thousands of iterations every time placement for a new graph is to be optimized [17].

## 2 Learning Method

The computation graph of a neural network can be modeled as a graph $G(V, E)$, where $V$ denotes the atomic computational operations (also referred to as "ops") in the neural network, and $E$ is the set of data communication edges. Each op $v \in V$ performs a specific computational function (e.g., convolution) on input tensors that it receives from its parent ops. For a set of devices $\mathcal{D} = \{d_1, \ldots, d_m\}$, a *placement* for $G$ is a mapping $\pi : V \to D$ that assigns a device to each op. The goal of device placement is to find a placement $\pi$ that minimizes $\rho(G, \pi)$, the duration of $G$'s execution

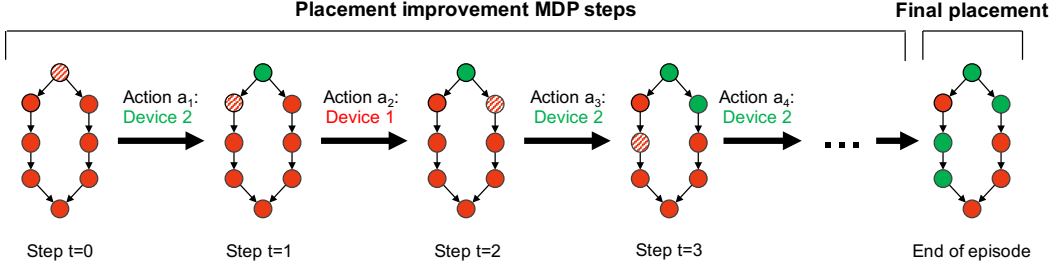

**Figure 1:** MDP structure of Placeto's device placement task. At each step, Placeto updates a placement for a node (shaded) in the computation graph. These incremental improvements amount to the final placement at the end of an MDP episode.

when its ops are placed according to $\pi$. To reduce the number of placement actions, we partition ops into predetermined *groups* and place ops from the same group on the same device, similar to Mirhoseini et.al. [12]. For ease of notation, henceforth we will use $G(V, E)$ to denote the graph of op groups. Here $V$ is the set of op groups and $E$ is set of data communication edges between op groups. An edge is drawn between two op groups if there exists a pair of ops, from the respective op groups, that have an edge between them in the neural network.

Placeto finds an efficient placement for a given input computation graph, by executing an iterative *placement improvement policy* on the graph. The policy is learned using RL over computation graphs that are structurally similar (i.e., coming from the same underlying probability distribution) as the input graph. In the following we present the key ideas of this learning procedure: the Markov decision process (MDP) formalism in §2.1, graph embedding and the neural network architecture for encoding the placement policy in §2.2, and the training/testing methodology in §2.3. We refer the reader to [22] for a primer on RL.

## 2.1   MDP Formulation

Let $\mathcal{G}$ be a family of computation graphs, for which we seek to learn an effective placement policy. We consider an MDP where a state observation $s$ comprises of a graph $G(V, E) \in \mathcal{G}$ with the following features on each node $v \in V$: (1) estimated run time of $v$, (2) total size of tensors output by $v$, (3) the current device placement of $v$, (4) a flag indicating whether $v$ has been "visited" before, and (5) a flag indicating whether $v$ is the "current" node for which the placement has to be updated. At the initial state $s_0$ for a graph $G(V, E)$, the nodes are assigned to devices arbitrarily, the visit flags are all $0$, and an arbitrary node is selected as the current node.

At a step $t$ in the MDP, the agent selects an action to update the placement for the current node $v$ in state $s_t$. The MDP then transitions to a new state $s_{t+1}$ in which $v$ is marked as visited, and an unvisited node is selected as the new current node. The episode ends in $n|V|$ steps, after the placement of each node has been updated $n$ times, where $n$ is a tunable hyper-parameter. Figure 1 illustrates this procedure for an example graph to be placed over two devices.

We consider two approaches for assigning rewards in the MDP: (1) assigning a zero reward at each intermediate step in the MDP, and a reward equal to the negative run time of the final placement at the terminal step; (2) assigning an intermediate reward of $r_t = \rho(s_t) - \rho(s_t + 1)$ at the $t$-th round for each $t = 0, 1, \ldots, n|V| - 1$, where $\rho(s)$ is the execution time of placement $s$. Intermediate rewards can help improve credit assignment in long training episodes and reduce variance of the policy gradient estimates [2, 15, 22]. However, training with intermediate rewards is more expensive, as it must determine the computation time for a placement at each step as opposed to once per episode. We contrast the benefits of either reward design through evaluations in Appendix A.4. To find a valid placement that fits without exceeding the memory limit on devices, we include a penalty in the reward proportional to the peak memory utilization if it crosses a certain threshold $M$ (details in Appendix A.7).

## 2.2   Policy Network Architecture

Placeto learns effective placement policies by directly parametrizing the MDP policy using a neural network, which is then trained using a standard policy-gradient algorithm [26]. At each step $t$ of the

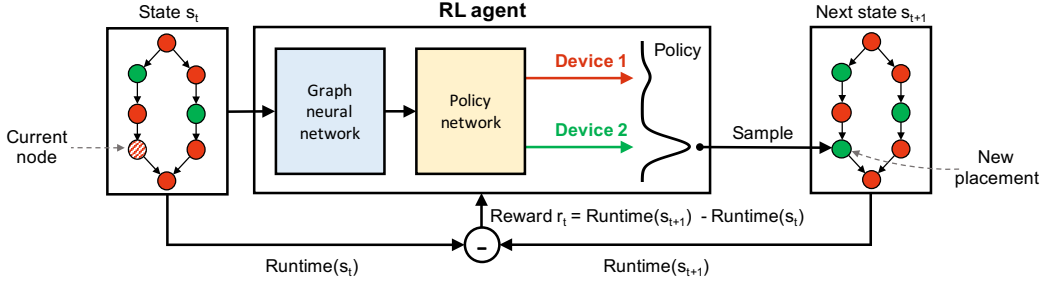

**Figure 2:** Placeto's RL framework for device placement. The state input to the agent is represented as a DAG with features (such as computation types, current placement) attached to each node. The agent uses a graph neural network to parse the input and uses a policy network to output a probability distribution over devices for the current node. The incremental reward is the difference between runtimes of consecutive placement plans.

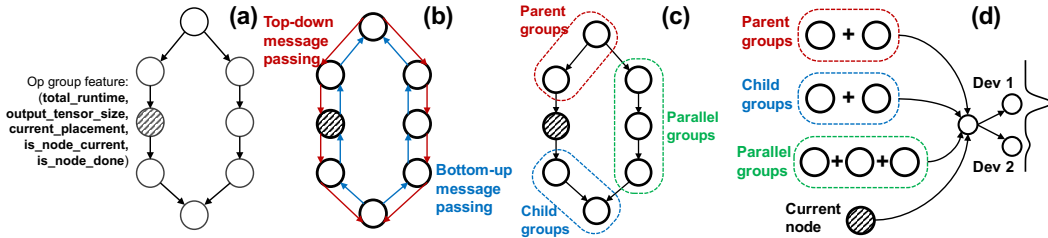

**Figure 3:** Placeto's graph embedding approach. It maps raw features associated with each op group to the device placement action. (a) Example computation graph of op groups. The shaded node is taking the current placement action. (b) Two-way message passing scheme applied to all nodes in the graph. Figure shows a single round of message passing. (c) Partitioning the message passed (denoted as bold) op groups. (d) Taking a placement action on two candidate devices for the current op group.

MDP, the policy network takes the graph configuration in state $s_t$ as input, and outputs an updated placement for the $t$-th node. However, to compute this placement action using a neural network, we need to first encode the graph-structured information of the state as a real-valued vector. Placeto achieves this vectorization via a *graph embedding* procedure, that is implemented using a specialized graph neural network and learned jointly with the policy. Figure 2 summarizes how node placements are updated during each round of an RL episode. Next, we describe Placeto's graph neural network.

**Graph embedding.** Recent works [3, 4, 8, 11] have proposed graph embedding techniques that have been shown to achieve state-of-the-art performance on a variety of graph tasks, such as node classification, link prediction, job scheduling etc. Moreover, the embedding produced by these methods are such that they can generalize (and scale) to unseen graphs. Inspired by this line of work, in Placeto we present a graph embedding architecture for processing the raw features associated with each node in the computation graph. Our embedding approach is customized for the placement problem and has the following three steps (Figure 3):

1. *Computing per-group attributes* (Figure 3a). As raw features for each op group, we use the total execution time of ops in the group, total size of their output tensors, a one-hot encoding of the device (e.g., device 1 or device 2) that the group is currently placed on, a binary flag indicating whether the current placement action is for this group, and a binary encoding of whether a placement action has already been made for the group. We collect the runtime of each op on each device from on-device measurements (we refer to Appendix 5 for details).

2. *Local neighborhood summarization* (Figure 3b). Using the raw features on each node, we perform a sequence of message passing steps [4, 8] to aggregate neighborhood information for each node. Letting $\mathbf{x}_v^{(i)}$ denote the node embeddings of op group $v$ after $i$ rounds of message passing, the updates take the form $\mathbf{x}_v^{(i+1)} \leftarrow g(\sum_{u \in \xi(v)} f(\mathbf{x}_u^{(i)}))$, where $\xi(v)$ is the set of neighbors of $v$, and $f, g$ are multilayer perceptrons with trainable parameters. We construct two directions (top-down from root groups and bottom-up from leaf groups) of message passings with separate parameters. The top-down messages summarize information about the subgraph of nodes that can reach $v$, while the bottom-up does so for the subgraph reachable from $v$. The parameters in the transformation functions $f, g$ are shared for message passing steps in each direction, among

all nodes. As we show in our experiments (§3), reusing the same message passing function everywhere provides a natural way to transfer the learned policy to unseen computation graphs. We repeat the message passing updates $k$ times, where $k$ is a tunable hyperparameter which can be set to sweep the entire graph or to integrate information from a local neighborhood of each node that is sufficient for making good placement decisions. In our experiments, we found that sweeping the entire graph is computationally expensive and provides little benefit compared to performing a fixed number (*e.g.,* $k = 8$) of iterations of message passing.

3. *Pooling summaries* (Figures 3c and 3d). After message passing, we aggregate the embeddings computed at each node to create a global summary of the entire graph. Specifically, for the node $v$ for which a placement decision has to be made, we perform three separate aggregations: on the set $S_{\text{parents}}(v)$ of nodes that can reach $v$, set $S_{\text{children}}(v)$ of nodes that are reachable by $v$, and set $S_{\text{parallel}}(v)$ of nodes that can neither reach nor be reached by $v$. On each set $S_i(v)$, we perform the aggregations using $h_i(\sum_{u \in S_i(v)} l_i(\mathbf{x}_u))$ where $\mathbf{x}_u$ are the node embeddings and $h_i, l_i$ are multilayer perceptrons with trainable parameters as above. Finally, node $v$'s embedding and the result from the three aggregations are concatenated as input to the subsequent policy network.

The above three steps define an end-to-end policy mapping from raw features associated with each op group to the device placement action.

## 2.3 Training

Placeto is trained using a standard policy-gradient algorithm [26], with a timestep-based baseline [7] (see Appendix A.1 for details). During each training episode, a graph from a set $\mathcal{G}_T$ of training graphs is sampled and used for performing the rollout. The neural network design of Placeto's graph embedding procedure and policy network allows the training parameters to be shared across episodes, regardless of the input graph type or size. This allows Placeto to learn placement policies that generalize well to unseen graphs during testing. We present further details on training in §3.

# 3 Experimental Setup

## 3.1 Dataset

We use Tensorflow to generate a computation graph given any neural network model, which can then be run to perform one step of stochastic gradient descent on a mini-batch of data. We evaluate our approach on computation graphs corresponding to the following three popular deep learning models: (1) **Inception-V3** [23], a widely used convolutional neural network which has been successfully applied to a large variety of computer vision tasks; (2) **NMT** [27], a language translation model that uses an LSTM based encoder-decoder and attention architecture for natural language translation; (3) **NASNet** [28], a computer vision model designed for image classification. For a more detailed descriptions of these models, we refer to Appendix A.2

We also evaluate on three synthetic datasets, each comprising of 32 graphs, spanning a wide range of graph sizes and structures. We refer to these datasets as *cifar10*, *ptb* and *nmt*. Graphs from *cifar10* and *ptb* datasets are synthesized using an automatic model design approach called ENAS [19]. The *nmt* dataset is constructed by varying the RNN length and batch size hyperparameters of the NMT model [27]. We randomly split these datasets for training and test purposes. Graphs in *cifar10* and *ptb* datasets are grouped to have about 128 nodes each, whereas graphs from *nmt* have 160 nodes. Further details on how these datasets are constructed can be found in the Appendix A.3.

## 3.2 Baselines

We compare Placeto against the following heuristics and baselines from prior work [13, 12, 6]:
(1) **Single GPU**, where all the ops in a model are placed on the same GPU. For graphs that can fit on a single device and don't have a significant inherent parallelism in their structure, this baseline can often lead to the fastest placement as it eliminates any cost of communication between devices.
(2) **Scotch** [18], a graph-partitioning-based static mapper that takes as input the computation graph, cost associated with each node, amount of data associated with connecting edges, and then outputs a placement which minimizes communication costs while keeping the load balanced across devices within a specified tolerance.
(3) **Human expert.** For NMT models, we place each LSTM layer on a separate device as recommended by Wu et al. [27]. We also colocate the attention and softmax layers with the final LSTM

layer. Similarly for vision models, we place each parallel branch on a different device.

(4) **RNN-based approach** [12], in which the placement problem is posed as finding a mapping from an input sequence of op-groups to its corresponding sequence of optimized device placements. An RNN model is used to learn this mapping. The RNN model has an encoder-decoder architecture with content-based attention mechanism. We use an open source implementation from Mirhoseini ,et.al. [12] available as part of the official Tensorflow repository [24]. We use the included hyperparameter settings and tune them extensively as required.

### 3.3 Training Details

**Co-location groups.** To decide which set of ops have to be co-located in an op-group, we follow the same strategy as described by Mirhoseini et al. [13] and use the final grouped graph as input to both Placeto and the RNN-based approach. We found that even after this grouping, there could still be a few operation groups with very small memory and compute costs left over. We eliminate such groups by iteratively merging them with their neighbors as detailed in Appendix A.6.

**Simulator.** Since it can take a long time to execute placements on real hardware and measure the elapsed time [12, 13], we built a reliable simulator that can quickly predict the runtime of any given placement for a given device configuration. We have discussed details about how the simulator works and its accuracy in Appendix A.5. This simulator is used only for training purposes. All the reported runtime improvements have been obtained by evaluating the learned placements on real hardware, unless explicitly specified otherwise.

Further details on training of Placeto and the RNN-based approach including our choice of hyperparameter values are given in the Appendix A.7. We open-source our implementation, datasets and the simulator. [2]

## 4 Results

In this section, we first evaluate the performance of Placeto and compare it with aforementioned baselines (§4.1). Then we evaluate Placeto's generalizability compared to the RNN-based approach (§4.2). Finally, we provide empirical validation for Placeto's design choices (§4.3).

### 4.1 Performance

Table 1 summarizes the performance of Placeto and baseline schemes for the Inception-V3, NMT and NASNet models. We quantify performance along two axes: (i) runtime of the best placement found, and (ii) time taken to find the best placement, measured in terms of the number of placement evaluations required for the RL-based schemes while training.

For all considered graphs, Placeto is able to rival or outperform the best comparing scheme. Placeto also finds optimized placements much faster than the RNN-based approach. For Inception on 2 GPUs, Placeto is able to find a placement that is $7.8\%$ faster than the expert placement. Additionally, it requires about $4.8\times$ fewer samples than the RNN-based approach. Similarly, for the NASNet model Placeto outperforms the RNN-based approach using up to $4.7\times$ fewer episodes.

For the NMT model with 2 GPUs, Placeto is able to optimize placements to the same extent as the RNN-based scheme, while using $3.5\times$ fewer samples. For NMT distributed across 4 GPUs, Placeto finds a non-trivial placement that is $16.5\%$ faster than the existing baselines. We visualize this placement in Figure 4. The expert placement heuristic for NMT fails to meet memory constraints of the GPU devices. This is because in an attempt to maximize parallelism, it places each layer on a different GPU, requiring copying over the outputs of the $i^{th}$ layer to the GPU hosting the $(i + 1)^{th}$ layer. These copies have to be retained until they can be fed in as inputs to the co-located gradient operations during the back-propagation phase. This results in a large memory footprint which ultimately leads to an OOM error. On the other hand, Placeto learns to exploit parallelism and minimize the inter-device communication overheads while remaining within memory constraints of all the devices. The above results show the advantage of Placeto's simpler policy representation: it is easier to learn a policy to incrementally improve placements, than to learn a policy that decides placements for all nodes in one shot.

| Model | Placement runtime (sec) | | | | | | Training time (# placements sampled) | | Improvement | |
|---|---|---|---|---|---|---|---|---|---|---|
| | CPU only | Single GPU | #GPUs | Expert | Scotch | Placeto | RNN-based | Placeto | RNN-based | Runtime Reduction | Speedup factor |
| Inception-V3 | 12.54 | 1.56 | 2 | 1.28 | 1.54 | 1.18 | 1.17 | 1.6 K | 7.8 K | - 0.85% | **4.8 ×** |
| | | | 4 | 1.15 | 1.74 | 1.13 | 1.19 | 5.8 K | 35.8 K | 5% | **6.1 ×** |
| NMT | 33.5 | OOM | 2 | OOM | OOM | 2.32 | 2.35 | 20.4 K | 73 K | 1.3 % | **3.5 ×** |
| | | | 4 | OOM | OOM | 2.63 | 3.15 | 94 K | 51.7 K | **16.5 %** | 0.55 × |
| NASNet | 37.5 | 1.28 | 2 | 0.86 | 1.28 | 0.86 | 0.89 | 3.5 K | 16.3 K | 3.4% | **4.7 ×** |
| | | | 4 | 0.84 | 1.22 | 0.74 | 0.76 | 29 K | 37 K | 2.6% | **1.3 ×** |

**Table 1:** Running times of placements found by Placeto compared with RNN-based approach [13], Scotch and human-expert baseline. The number of measurements needed to find the best placements for Placeto and the RNN-based are also shown (K stands for kilo). Reported runtimes are measured on real hardware. Runtime reductions and speedup factors are calculated with respect to the RNN-based approach. Lower runtimes and lower training times are better. OOM: Out of Memory. For NMT model, the number of LSTM layers is chosen based on the number of GPUs.

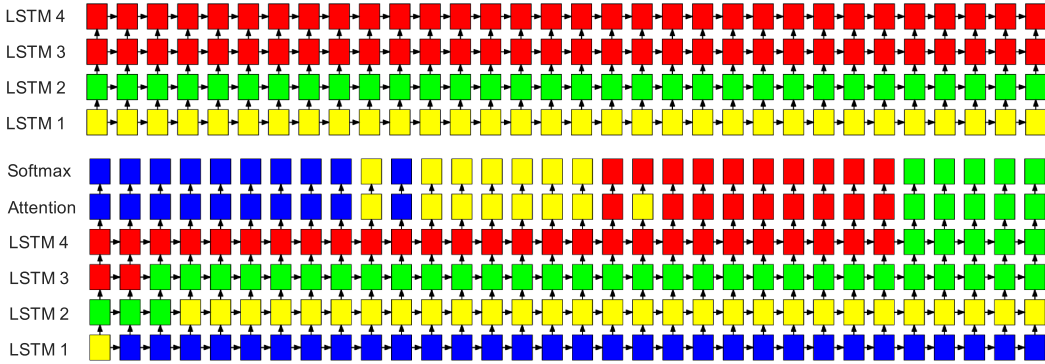

**Figure 4:** Optimized placement across 4 GPUs for a 4-layered NMT model with attention found by Placeto. The top LSTM layers correspond to encoder and the bottom layers to decoder. All the layers are unrolled to a maximum sequence length of 32. Each color represents a different GPU. This non-trivial placement meets the memory constraints of the GPUs unlike the expert-based placement and the Scotch heuristic, which result in an Out of Memory (OOM) error. It also runs 16.5% faster than the one found by the RNN-based approach.

## 4.2 Generalizability

We evaluate generalizability of the learning-based schemes, by training them over samples of graphs drawn from a specific distribution, and using the learned policies to predict effective placements for unseen graphs from the same distribution during test time.

If the placements predicted by a policy are as good as placements found by separate optimizations over the individual test graphs, we conclude that the placement scheme generalizes well. Such a policy can then be applied to a wide variety of structurally-similar graphs without requiring re-training. We consider three datasets of graphs called *nmt, ptb* and *cifar10* for this experiment.

For each test graph in a dataset, we compare placements generated by the following schemes: *(1) Placeto Zero-Shot.* A Placeto policy trained over graphs from the dataset, and used to predict placements for the test graph without any further re-training. *(2) Placeto Optimized.* A Placeto policy trained specifically over the test graph to find an effective placement. *(3) Random.* A simple strawman policy that generates placement for each node by sampling from a uniform random distribution. We define *RNN Zero-Shot* and *RNN Optimized* in a similar manner for the RNN-based approach.

Figure 5 shows CDFs of runtimes of the placements generated by the above-defined schemes for test graphs from *nmt, ptb* and *cifar10* datasets. We see that the runtimes of the placements generated by *Placeto Zero-Shot* are very close to those generated by *Placeto Optimized*. Due to Placeto's generalizability-first design, *Placeto Zero-Shot* avoids the significant overhead incurred by *Placeto Optimized* and *RNN Optimized* approaches, which search through several thousands of placements before finding a good one.

Figure 5 also shows that *RNN Zero-Shot* performs significantly worse than *RNN Optimized*. In fact, its performance is very similar to that of *Random*. When trained on a graph, the RNN-based

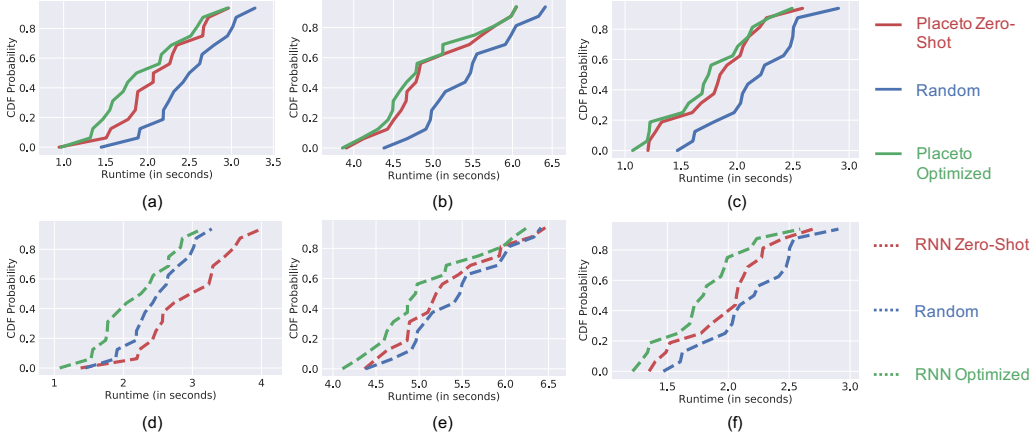

**Figure 5:** CDFs of runtime of placements found by the different schemes for test graphs from (a), (d) *nmt* (b), (e) *ptb* and (c), (f) *cifar10* datasets. Top row of figures ((a), (b), (c)) correspond to Placeto and bottom row ((d), (e), (f)) to RNN-based approach. *Placeto Zero-Shot* performs almost on par with fully optimized schemes like *Placeto Optimized* and *RNN Optimized* even without any re-training. In contrast, *RNN Zero-Shot* performs much worse and only slightly better than a randomly initialized policy used in *Random* scheme.

approach learns a policy to search for an effective placement for that graph. However, this learned search strategy is closely tied to the assignment of node indices and the traversal order of the nodes in the graph, which are arbitrary and have a meaning only within the context of that specific graph. As a result, the learned policy cannot be applied to graphs with a different structure or even to the same graph using a different assignment of node indices or traversal order.

## 4.3 Placeto Deep Dive

In this section we evaluate how the node traversal order of a graph during training, affects the policy learned by the different learning schemes. We also present an ablation study of Placeto's policy network architecture. In Appendix A.4 we conduct a similar study on the benefits of providing intermediate rewards in Placeto during training.

**Node traversal order.** Unlike the RNN-based approach, Placeto's use of a graph neural network eliminates the need to assign arbitrary indices to nodes while embedding the graph features. This aids in Placeto's generalizability, and allows it to learn effective policies that are not tied to the specific node traversal orders seen during training. To verify this claim, we train Placeto and the RNN-based approach on the Inception-V3 model following one of 64 fixed node traversal orders at each episode. We then use the learned policies to predict placements under 64 unseen random node traversal orders for the same model. With Placeto, we observe that the predicted placements have runtimes within 5% of that of the optimized placement on average, with a difference of about 10% between the fastest and slowest placements. However, the RNN-based approach predicts placements that are about 30% worse on average.

**Alternative policy architectures.** To highlight the role of Placeto's graph neural network architecture (§2.2), we consider the following two alternative policy architectures and compare their generalizability performance against Placeto's on the *nmt* dataset.
*(1) Simple aggregator*, in which a feed-forward network is used to aggregate all the node features of the input graph, which is then fed to another feed-forward network with softmax output units for predicting a placement. This simple aggregator performs very poorly, with its predicted placements on the test dataset about 20% worse on average compared to Placeto.
*(2) Simple partitioner*, in which the node features corresponding to the parent, child and parallel nodes—of the node for which a decision is to be made—are aggregated independently by three different feed-forward networks. Their outputs are then fed to a separate feed-forward network with softmax output units as in the simple aggregator. Note that this is similar to Placeto's policy architecture (§2.2), except for the local neighborhood summarization step (i.e., step 2 in §2.2). This results in the simple partitioner predicting placements that run 13% slower on average compared to Placeto. Thus, local neighborhood aggregation and pooling summaries from parent, children and

parallel nodes are both essential steps for transforming raw node features into generalizable embeddings in Placeto.

## 5   Conclusion

We presented Placeto, an RL-based approach for finding device placements to minimize training time of deep-learning models. By structuring the policy decisions as incremental placement improvement steps, and using graph embeddings to encode graph structure, Placeto is able to train efficiently and learns policies that generalize to unseen graphs.

Placeto currently relies on a manual grouping procedure to consolidate ops into groups. An interesting future direction would be to extend our approach to work directly on large-scale graphs without manual grouping [17] [12]. Another interesting future direction would be to learn to make placement decisions that exploit both model and data parallelism which can lead to a significant increase in the margin of runtime improvements possible [9] [20].

**Acknowledgements.** We thank the anonymous NeurIPS reviewers for their feedback. This work was funded in part by NSF grants CNS-1751009, CNS-1617702, a Google Faculty Research Award, an AWS Machine Learning Research Award, a Cisco Research Center Award, an Alfred P. Sloan Research Fellowship, and the sponsors of MIT Data Systems and AI Lab. We also gratefully acknowledge Cloudlab [5] and Chameleon testbeds [10] for providing us with compute environments for some of the experiments.

## Footnotes

[2]https://github.com/aravic/generalizable-device-placement

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
