[Supplementary Material]

# Appendices

## A   Implementation Details

### A.1   REINFORCE Algorithm

Placeto is trained using the REINFORCE policy-gradient algorithm [24], in which a Monte-Carlo estimate of the gradient is used for updating policy parameters. During each training episode, a graph $G$ is sampled from the set of training graphs $\mathcal{G}_T$ (see §2.1) and a rollout $(s_t, a_t, r_t)_{t=0}^{N-1}$ is performed on $G$ using the current policy $\pi_\theta$. Here $s_t, a_t, r_t$ refer to the state, action and reward at time-step $t$ respectively, and $\theta$ is the parameter vector encoding the policy. At the end of each episode, the policy parameter $\theta$ is updated as

$$\theta \leftarrow \theta + \eta \sum_{i=0}^{N-1} \nabla_\theta \log \pi_\theta(a_i|s_i) \left( \sum_{i'=i}^{N-1} r_{i'} - b_i \right), \tag{1}$$

where $b_i$ is a baseline for reducing variance of the estimate, and $\eta$ is a learning rate hyperparameter. Placeto uses a time-based baseline in which $b_i$ is computed as the average of cumulative rewards $\sum_{i'=i}^{N-1} r_t$ at time-step $i$ over multiple independent rollouts of graph $G$ using the current policy $\pi_\theta$. Intuitively, the update rule in Equation (1) shifts $\theta$ such that the probability of making "good" placement actions (i.e., actions for which the cumulative rewards are higher than the average reward) is increased and vice-versa. Thus over the course of training, Placeto gradually learns placement policies for which the overall running time of graphs, coming from the same distribution as $\mathcal{G}_T$, are minimized.

### A.2   Models

We evaluate our approach on the following popular deep-learning models from Computer Vision and NLP tasks:

1. **Inception-V3** [21] is a widely used convolutional neural network which has been successfully applied to a large variety of computer vision tasks. Its network consists of a chain of blocks, each of which has multiple branches made up of convolutional and pooling operations. While these branches from a block can be executed in parallel, each block has a sequential data dependency on its predecessor. The network's input is a batch of 64 images each with dimension $299 \times 299 \times 3$. Its computational graph in tensorflow has 3002 operations.

2. **NMT** [25] Neural Machine Translation with attention is a language translation model that uses an LSTM based encoder-decoder architecture to translate a source sequence into a target sequence. When its computational graph is unrolled to handle input sequences of length up to 32, the memory footprint to hold the LSTM hidden states can be large, potentiating the use of model parallelism. We consider 2-layer as well as 4-layer versions depending on the number of GPUs available for placement. Their computational graphs in tensorflow have 6361 and 10812 operations respectively. We use a batch size of 128.

3. **Nasnet** [26] is a computer vision model designed for image classification. Its network consists of a series of cells each of which has multiple branches of computations that are finally reduced at the end to form input for the next cell. It's computational graph consists of 12942 operations. We use a batch size of 64.

Prior works [11, 10, 5] report significant possibilities of improvements in runtimes for several of the above models when placed over multiple GPUs.

### A.3   Datasets

We evaluate the generalizability of each placement scheme by measuring how well it transfers a placement policy learned using the graphs from a training dataset to unseen graphs from a test dataset.

To our knowledge, there is no available compilation of tensorflow models that is suitable to be used as a training dataset for the device placement problem. For example, one of the most popular tensorflow model collection called ONNX [14] has only a handful of models and most of them do not have any inherent model parallelism in their computational graph structure.

To overcome this difficulty, we use an automatic model design approach called ENAS [17] to generate a variety of neural network architectures of different shapes and sizes. ENAS uses a Reinforcement learning-based controller to discover neural network architectures by searching for an optimal subgraph within a larger graph. It is trained to maximize expected reward on a validation set.

We use the classification accuracy on CIFAR-10 dataset as a reward signal to the controller so that over the course of its training, it generates several neural network architectures which are designed to achieve high accuracy on the CIFAR-10 image classification task.

We randomly sample from these architectures to form a family of $N$ tensorflow graphs which we refer to as the *cifar-10* dataset. Furthermore, for each of these graphs, batch size is chosen by uniformly sampling from the interval, $bs_{low}$ to $bs_{high}$ creating a range of memory requirements for the resulting graphs. We use a fraction $f$ of these graphs for training and the remaining for testing.

Similar to the *cifar-10* dataset, we use the inverse of validation perplexity [23] on Penn Treebank dataset as a reward signal to generate a class of tensorflow graphs suitable for language modeling task which we refer to as the *ptb* dataset. Furthermore, we also vary the number of unrolled steps $L$ for the recurrent cell by sampling uniformly from $L_{low}$ to $L_{high}$.

In addition to the above two datasets created using the ENAS method, we create a third dataset made of graphs based on the NMT model which we refer to as the *nmt* dataset. We generate $N$ different variations of the 2-layer NMT model by sampling the number of unrolled steps, $L$ from $L_{low}$ to $L_{high}$ and batch size from $bs_{low}$ to $bs_{high}$. This creates a range of complex graphs based on the common encoder-decoder with attention structure with a wide range of memory requirements.

For our experiments, we use the following settings: $N = 32$, $f = 0.5$, $bs_{low} = 240$, $bs_{high} = 360$ for *cifar10* graph dataset, $N = 32$, $f = 0.5$, $bs_{low} = 1536$, $bs_{high} = 3072$, $L_{low} = 25$, $L_{high} = 40$ for *ptb* dataset and $N = 32$, $f = 0.5$, $bs_{low} = 64$, $bs_{high} = 128$, $L_{low} = 16$, $L_{high} = 32$ for *nmt* dataset.

We open-source these datasets.[3] We visualize some samples graphs from *cifar-10* and *ptb* datasets in Figures 6 and 7

## A.4 Intermediate Rewards

Placeto's MDP reformulation allows us to provide intermediate reward signals that are known to help with the temporal credit assignment problem.

Figure 8 empirically shows the benefits of having intermediate rewards as opposed to a single reward at the end. They lead to a faster convergence and a lower variance in cumulative episodic reward terms used by REINFORCE to estimate policy gradients during training. Placeto's policy network learns to incrementally generate the whole placement through iterative improvement steps over the course of the episode starting from a trivial placement.

## A.5 Simulator

Over the course of training, runtimes for thousands of sampled placements need to be determined before a policy can be trained to converge to a good placement. Since it is costly to execute the placements on real hardware and measure the elapsed time for one batch of gradient descent [10, 11], we built a simulator that can quickly predict the runtime of any given placement for a given device configuration.

For any given model to place, our simulator first profiles each operation in its computational graph by measuring the time it takes to run it on all the available devices. We model the communication cost between devices as linearly proportional to the size of intermediate data flow across operations.

The simulator maintains the following two FIFO queues for each device $d$:

- $Q_d^{\mathrm{op}}$: Collection of operations that are ready to run on $d$.
- $Q_d^{\mathrm{transfer}}$: Collection of output tensors that are ready to be transferred from $d$ to some other device.

We deem an operation to be runnable on a device $d$ only after all of its parent operations have finished executing and their corresponding output tensors have been transferred to $d$.

Our simulator uses an event-based design to generate an execution timeline. Each event has a timestamp at which it gets triggered. Further, it also includes some metadata for easy referencing. We define the following types of events:

- *Op-done*: Used to indicate when an operation has finished executing. Its timestamp is determined based on the information collected from the initial profiling step on how long it takes to run the operation on its corresponding device.
- *Transfer-done*: Used to indicate the finish of an inter-device transfer of an output tensor. Its timestamp is determined using an estimated communication bandwidth $b$ between devices and size of the tensor.

**Figure 6:** Sample graphs from *cifar-10* dataset. Each color represents a different GPU in the optimized placement. Size of the node indicates its compute cost and the edge thickness visualizes the communication cost. The above graphs exhibit a wide range of structure and connectivity.

**Figure 7:** Few of the recurrent cells used to generate the sequence based models in *ptb* dataset. Each color indicates a different operation from the following: Identity (Id), Sigmoid (Sig.), Tanh (tanh), ReLU (ReLU). $x[t]$ is the input to the cell and the final add operation is its output.

**Figure 8:** (a) Cumulative Episodic rewards when Placeto is trained with and without intermediate rewards on NMT model. Having intermediate rewards within an episode as opposed to a single reward at the end leads to a lower variance in the runtime. (b) Runtime improvement observed in the final episode starting from the initial trivial placement.

- *Wakeup*: Used to signal the wakeup of a device (or a bus) that has been marked as free after its operation queue (or transfer queue) became empty and there was no pending work for it to do.

We now define event handlers for each of the above event-types

**Op-done event-handler:**

Whenever an operation $o$ has completed running on the device $d$, the simulator performs the following actions in order:

- For every child operation $o'$ placed on device $d'$:
    - Enqueue output tensor $t_o$ of $o$ to $Q_d^{\text{transfer}}$ if $d \neq d'$.
    - Check if $o'$ is runnable. If so, enqueue it to $Q_{d'}^{\text{op}}$.
    - Add the appropriate *Wakeup* events necessary after the above two steps in case $d'$ happens to be free.
- If $Q_d^{\text{op}}$ is empty, then mark the device $d$ as free. Otherwise, pick the next operation from this queue and create its corresponding *Op-done* event.

**Transfer-done event-handler:**

Whenever a tensor $t$ has been transferred from device $d$ to $d'$, the simulator performs following actions in order:

- Check if the operation $o$ on device $d'$ that takes $t$ as its input is runnable. If so, enqueue it to $Q_{d'}^{\text{op}}$. Add a *Wakeup* event for device $d'$ if necessary.
- If $Q_d^{\text{transfer}}$ is empty, mark the bus corresponding to device $d$ as free. Otherwise, pick the next transfer operation and create its corresponding *Op-done* event.

**Wakeup event-handler:** If a device or its corresponding bus receives a *wakeup* signal, then its corresponding queue should be non-empty. Pick the first element from this queue and create a new *Op-done* or *Transfer-done* event based on it.

We initialize the queues with operations that have no data-dependencies and create their corresponding *Op-done* events. The simulation ends when there are no more events left to process and all the operations have finished executing. The timestamp on the last *Op-done* event is considered to be the simulated runtime.

During simulation, we keep track of the start and end timestamps for each operation. Along with the tensor sizes, these are used to predict the peak memory-utilizations of the devices.

Note that we've tried to model our simulator based on the real execution engine used in Tensorflow. We've validated that the following key aspects of our design match with tensorflow's implementation: (a) Per-device FIFO queues holding runnable operations. (b) Communication overlapping with compute. (c) No more than one operation runs on a device at a time.

As a result, an RL-based scheme trained with the simulator exhibits nearly identical run times compared to training directly on the actual system. We demonstrate this by comparing the run times in the learning curves of a RNN-based approach [11] on the real hardware and our simulator (Figure 9).

**Figure 9:** RNN-based approach exhibits near identical learning curve when reward signal is from a simulator or directly from measurements on real machines.

## A.6 Merge-and-Colocate heuristic

Merge-and-Colocate is a simple heuristic designed to reduce the size of a graph by colocating small operations with their neighbors.

Given any input graph $G_i$, the Merge-and-Colocate heuristic first merges the node with the lowest cost into its neighbor. If the node has no neighbors, then its predecessor is used instead. This step is repeated until the graph size reaches a desired value $N$ or alternatively until there are no more nodes with cost below a certain threshold $C$. The merged nodes are then colocated together on to the same device. For our experiments, we use the size of the output tensor of an operation as the cost metric for the above proceAdure.

## A.7 Training details

Here, we describe training details for Placeto and RNN-based model. Unless otherwise specified, we use the same described methodology for setting the hyper-parameters for both of these approaches.

**MDP.** Given any graph with $|V|$ nodes, our episode ends in $n|V|$ steps, after the placement of each node has been updated n times. We find that using a value of $n = 1$ or $n = 2$ strikes a good balance between performance and speed.

**Entropy.** We add an entropy term in the loss function as a way to encourage exploration. We tune the entropy factor seperately for Placeto and RNN-based model so that the exploration starts off high and decays gradually to a low value towards the final training episodes.

**Optimization.** We tune the initial learning rate for each of the models that we report the results on. For each model, we decay the learning rate linearly to smooth convergence. We use Adam's optimizer to update our policy weights. We found that using an initial learning rate between $10^{-3}$ and $10^{-4}$ yields best results.

**Workers.** We use 8 worker threads and a master coordinator which also serves as a parameter server. At the beginning of every episode, each worker synchronizes its policy weights with the parameter server. Each worker then independently performs an episode rollout and collects the rewards for its sampled placement. It then computes the gradients of reinforce loss function with respect to all the policy parameters. All the workers send their respective gradients to the parameter server which sums them up and updates the parameters to be used for the next episode.

**Baselines.** For Placeto, we use a seperate moving average baseline for each stage of the episode. The baseline for time step $t$ is the average of cumulative rewards at step $t$, of the past $k$ episodes where $k$ is a tunable hyperparameter. We use $k = 10$ in most of our experiments.

For RNN-based approach, we use baseline as described in Mirhoseini et al. [11].

**Neural Network Architecture** For Placeto, we use feed-forward networks during message passing and aggregation steps. Each of these feed-forward networks is made up of two hidden layers each with $d = 8$ hidden units. We use $|E| = 8$ for the dimension of the node embeddings. We feed the outputs of the aggregator into a two layer feed-forward neural network with $d = 16$ hidden units and a softmax output layer. We use ReLU as our default activation function.

We repeat the message passing updates $k$ times, where k is a tunable hyper-parameter which can be set to sweep the entire graph or to integrate information from a local neighborhood of each node that is sufficient for making good placement decisions. We found that using a value of $k = 4$ or $k = 8$ works best in practice.

For the RNN-based approach, we use a bi-directional RNN with a hidden size of $512$ units.

**Training Details:** We use distributed learning with synchronous SGD algorithm to train Placeto's policy network. A parameter server is used to co-ordinate updates with 8 worker nodes. Each worker independently performs an episode rollout and collects the rewards for its sampled placement. It then computes the gradients of reinforce loss function with respect to all the policy parameters. All the workers then send their respective

gradients to the parameter server which sums them up before updating the parameters to be used for the next episode. To train a policy using multiple graphs, a different graph is used by each worker.

The stopping criterion used to determine the training time in Table 1 is the numerical convergence of reward signal over a long enough measurement window. Precisely, the reward value doesn't change more than 2% in a measurement window of 1000 episodes

**Reward:**

Given any placement $p$ with runtime $r$ (in seconds) and maximum peak memory utilization $m$ (in GB) across all devices, we define memory penalized runtime, $R(p)$ as follows:

$$R(p) = r + c * ReLU(m - M)$$

where $M$ is the total available memory on the device with maximum peak memory utilization and $c$ is a scale factor. For our experiments, we use $c = 2$.

To find a valid placement that fits without exceeding the memory limit on devices, we include a penalty proportional to the peak memory utilization if it crosses a certain threshold $M$. This threshold $M$ could be used to control the memory footprint of the placements under execution environments with high memory pressure (*e.g.,* GPUs). For instance, we use $M = 10.7$ GB in our experiments to find placements that fit on Tesla K80 GPUs which have about 11 GB of memory available for use.

For an MDP episode of length T, we propose the following two different ways to assign reward:

- **Terminal Reward:** A Non-zero reward is given only at the end of the episode. That is, $r_1 = 0, r_2 = 0, \ldots r_T = -R(p_T)$. This requires evaluating only one placement per episode but leads to a high variance in the policy gradient estimates due to a lack of temporal credit assignment.

- **Intermediate Rewards:** Under this setting, the improvement in runtimes of the successive time steps of an episode is used as an intermediate reward signal. That is, $r_1 = R(p_1) - R(p_0)$, $r_2 = R(p_2) - R(p_1)$, $\ldots$ $r_T = R(p_T) - R(p_{T-1})$. Although this requires evaluating $T+1$ placements for every episode, intermediate rewards result in better convergence properties in RL [24].

**Devices:**

We target the following device configuration for optimizing placements: Tensorflow r1.9 version running on a p2.8xlarge instance from AWS EC2 [1] equipped with 8 Nvidia Tesla K80 GPUs and a Xeon E5-2686 broadwell processor.

## Footnotes

[3]https://github.com/aravic/generalizable-device-placement/tree/master/datasets