[Reviews · NeurIPS 2019]

Reviewer 1



This paper proposed a policy based approach to device placement. It starts with an i initialized placement and the policy learns to update the placement of a candidate node by optimizing the delta in reward function (runtime). The main advantage of this work over prior methods is generalization, which is a key requirement for adopting this method in real-world systems. The mutating approach along with the use of graph neural networks seem to improve the sample efficiency and robustness of results. It is very clearly written and experiments show significant improvements over baselines.

Reviewer 2



Originality: The use of graph neural networks appears novel (concurrent with Paliwal), as does the sweep order (for which I don't know other papers, at least for this application of graph neural networks). The trick of using architecture search as a dataset also seems novel, and I'm quite happy with this idea. Quality: The submission is sound, but I have a few minor concerns: 1. Why use REINFORCE, rather than a better RL algorithm such as PPO (as in Spotlight)? It's possible REINFORCE is good enough, but I'm skeptical given that (1) REINFORCE is much worse in normal RL environments and (2) the paper explicitly presents evidence that using an incremental baseline helps learning. The learned value function in PPO, Q-learning, etc. could potentially play the same variance reduction role or even do quite a lot better (presumably not all of the variance due to upstream moves is explained by reward so far). 2. The node traversal order ablation is a bit silly: it doesn't feel very interesting whether we can sabotage training order and not lose accuracy for other orders. I was specifically disappointed because I was hoping to see an ablation for the bidirectional sweep approach vs. updating every node at once as with standard GNNs. Clarity: I found several aspects of the paper difficult to parse, or even slightly misleading. I think these are all fixable during the review process. 1. As mentioned above, the authors claim as a contribution iterative improvement of policies. The architecture they train supports being used in this way, by sweeping over the graph multiple times with each vertex flagged for change multiple times. However, the paper says "The episode ends in |V| steps when the placements for all the nodes have been updated." This is the same number of steps as the RNN approach (though the RNN approach does much less work per step). It's still possible that at test time they do more than |V| steps at test time, but I don't see any mention of this. To be explicit: I don't think it is reasonable to say the RNN approach outputs a whole placement in one step, since the placement of future nodes can condition on the sampled placement of previous nodes in exactly the same way as this paper. 2. Given that the architecture *does* support multiple epochs of placement and replacement, some discussion seems in order for why this isn't done. 3. What's k (the number of graph sweeps performed per step)? What are the details of the Placeto network, such as the embedding size? 4. I'm somewhat surprised that sweeps through a deep graph would work using (as far as I can tell from the paper) pure feed forward networks at message passing nodes. At least early GNN papers used LSTMs / GRUs / similar at nodes, but both this paper and Paliwal just use MLPs. I'd be curious for discussion of this, or a pointer to a reference that explains the switch if a good one exists. 5. I'm confused about the "Training time" column in Table 1. Is this for Placeto optimized, such that the number of steps is RL training on the specific graph we're trying to optimize (ignoring prior training on the ensemble of different graphs)? If so, what's the stopping condition? 6. Minor cleanup: in the supplementary material, the reward can be written more concisely as R(p) = r + c relu(m − M), which also highlights the continuity. Significance: I think it's quite likely these results will be built on, in particular the use of graph neural networks for both device placement and other computation graphs applications, and the use of neural architecture search outputs as a dataset.

Reviewer 3



The paper presents a new approach to device placement of a compute graph by using a graph embedding neural network. The graph embedding network computes features that are used in the a policy to compute which device the next node in the graph should be assigned to (It appears to me, that the order of the nodes is possibly fixed and not addressed by the policy; the order seems to be based on how the graph is fed into the model, and the authors make the point of showing that the method is more resilient to changes in order of nodes at test time, compared to RNNs in section 4). The learning algorithm used is a policy gradient algorithm with time step based baselines. The paper presents the ideas very clearly, and has nice results. The contribution is more from the combination of ideas than from a new idea. Graph embedding networks have been applied for various tasks, and device placement has been addressed previously by Mirhoseni et al. However, this paper shows how to combined these thoughts, allowing policies that are resilient to changes in node order and also generalizes to other graphs.

[Author Response · NeurIPS 2019]

We would like to sincerely thank all our reviewers for their valuable feedback and insightful comments. We are committed to doing our best in addressing their concerns and suggestions in the final version of the paper. We would like to reiterate that apart from making the suggested clarifying revisions to our paper, we will release all of the datasets, source code and hyper-parameter configurations used to ease the reproducibility of our work.

**Formal meaning of the word "family" in Section 4.2 (Review #1):** In experiments on generalizability, we've used the term "family" in lieu of a more formal choice: "distribution". More precisely, we're interested in evaluating whether a placement policy trained with graphs sampled from a particular distribution can generalize to different, unseen graphs sampled from the same distribution. In our evaluation, we pick these distributions to correspond to neural network architectures designed for a specific task, such as image recognition (*cifar-10*), language modeling (*ptb*) and language translation (*nmt*). Appendix A.3 describes how we generated neural networks for each of these tasks using the ENAS method for *cifar-10* and *ptb*, and variations of the NMT model for *nmt*. We will clarify this point in the revision.

**Choice of REINFORCE algorithm for RL optimization (Review #2):** We've chosen REINFORCE for RL optimization in order to establish a fair comparison to the prior work, a majority of which uses it as well (Mirhoseini *et al.* (2017), Mirhoseini *et al.* (2018) and Paliwal *et al.* (2019)). Furthermore, we believe that our core contributions are orthogonal to any specific choice of the optimization algorithm used. However, we do agree that it would be interesting to verify if our results improve with the use of a better algorithm that plays a more significant role in variance reduction. We leave this investigation for future work.

**Graph Neural Network architecture (Review #2):** Placeto's policy is computed by running two separate, independent graph neural networks concurrently and concatenating node embeddings obtained from them together to form embeddings that get used for placement decisions. The first graph neural network iteratively aggregates messages for $k$ steps "bottom-up" from the children nodes using the functions $f$ and $g$ as follows: $\mathbf{x}_v^{(i+1)} \leftarrow g(\sum_{u \in \xi(v)} f(\mathbf{x}_u^{(i)}))$, where $\xi(v)$ are the children of node $v$. The second graph neural network performs a similar update "top-down" from the parent nodes instead. $k$ is a tunable hyperparameter, which can be set to sweep the entire graph or to integrate information from a local neighborhood of each node that is sufficient for making good placement decisions. In our experiments, we found that sweeping the entire graph is computationally expensive and provides little benefit compared to $k = 8$. Hence we use $k = 8$ in all of our experiments. We will revise Section 2.2 and Figure 3 to clarify this point. Regarding the use of feed-forward neural networks in the message passing operations of the GNN, our architecture is similar to past work (*e.g.,* Mao *et al.* (2018), Battaglia *et al.* (2018)).

*Additional details:* We use a graph embedding size of $E = 8$. All our feed-forward networks have a single hidden layer. We will add these details to the appendix section A.7 and additionally, release a source file with all of of our hyper-parameter configurations along with the source code for easy reproducibility of our results.

**Ablation study of bidirectional GNN (Review #2):** We agree that it would make for an interesting addition to compare our bi-directional approach to a more standard one where messages from all the neighbors (parents and children nodes) are aggregated jointly at each step of the message passing update operation. We plan on performing this ablation study and including its results in the final version of our paper.

**Clarification on iterative placement improvement vs. RNN (Review #2):** While the RNN approach could indeed condition placement of future nodes on those chosen for prior nodes, our formulation makes this conditioning explicit in the state space of the MDP, since the state includes the current placements as a feature associated with each node. By contrast, prior RNN-based architectures (e.g., the Encoder/Decoder approach in Mirhoseini et al.) must realize such conditioning implicitly via the hidden state of the RNN. As the reviewer observes, making this conditioning explicit also enables us to perform multiple epochs of placement improvements. Although we failed to mention this, we actually do perform multiple epochs of iterative placement improvement in our experiments. More specifically, we found that performing 2 rounds of epochs strikes a good balance between performance and speed. We will revise Section 2.1 to explain this point precisely. In particular, we will modify the MDP description to state that the episode ends in $n|V|$ steps, where $n$ is the number of epochs.

**Stopping criterion used for Section 4.1 (Review #2):** The stopping criterion used to determine the training time column in Table 1 is the numerical convergence of reward signal over a long enough measurement window. Precisely, the reward value doesn't change more than 2% in a measurement window of 1000 episodes.

**More succinct choice of reward formulation (Review #2):** Thank you for the suggestion! We will amend the reward section in the Appendix A.7 to reflect this more succinct $ReLU$-based expression.

**Node ordering used for training (Review #3):** We pick an arbitrary topological ordering (in which parents nodes appear before their children) by default. This ordering is indeed not learned. However, as we show in Section 4.3, the order does not impact Placeto, and can even be completely random (not topological) without having a significant impact on generalizability of the learned policy for Placeto.

[Meta-Review · NeurIPS 2019]

The paper introduces a new RL-based approach to device placement in computation graphs that relies on using graph embedding neural network instead of RNNs. The reviewers were all impressed by the novelty of the proposed approach, the significance of the empirical results, as well as by the ability of the method to generalize across different tasks. While preparing the final version, please take into account the detailed comments and suggestions mentioned in the reviews.